# Imaging of Oligometastatic Disease

**DOI:** 10.3390/cancers14061427

**Published:** 2022-03-10

**Authors:** Naik Vietti Violi, Rami Hajri, Laura Haefliger, Marie Nicod-Lalonde, Nicolas Villard, Clarisse Dromain

**Affiliations:** 1Department of Radiology, Lausanne University Hospital (CHUV), Rue du Bugnon 46, 1011 Lausanne, Switzerland; naik.vietti-violi@chuv.ch (N.V.V.); rami.hajri@chuv.ch (R.H.); laura.haefliger@chuv.ch (L.H.); nicolas.villard@chuv.ch (N.V.); 2Department of Nuclear Medicine, Lausanne University Hospital (CHUV), Rue du Bugnon 46, 1011 Lausanne, Switzerland; marie.nicod-lalonde@chuv.ch

**Keywords:** oligometastatic disease, imaging

## Abstract

**Simple Summary:**

The imaging of oligometastatic disease (OMD) is challenging as it requires precise loco-regional staging and whole-body assessment. The combination of imaging modalities is often required. The more accurate imaging tool will be selected according to tumor type, the timing with regard to measurement and treatment, metastatic location, and the patient’s individual risk for metastasis. The most commonly used modalities are contrast-enhanced computed tomography (CT), magnetic resonance imaging and metabolic and receptor-specific imaging, particularly, ^18^F-fluorodesoxyglucose positron emission tomography/CT, used alone or in combination.

**Abstract:**

Oligometastatic disease (OMD) is an emerging state of disease with limited metastatic tumor burden. It should be distinguished from polymetastatic disease due the potential curative therapeutic options of OMD. Imaging plays a pivotal role in the diagnosis and follow-up of patients with OMD. The imaging tools needed in the case of OMD will differ according to different parameters, which include primary tumor type, timing between measurement and treatment, potential metastatic location and the patient’s individual risk for metastasis. In this article, OMD is defined and the use of different imaging modalities in several oncologic situations are described in order to better understand OMD and its specific implication for radiologists.

## 1. Introduction

Oligometastatic disease (OMD) was first described by Hellman et al. in 1995 [1] as an intermediate stage between localized tumor and widely polymetastatic disease (PMD). This first definition was mainly based on the number of metastatic lesions, which ranged from one to five. In parallel, the management paradigm of patients with metastatic disease changed from a palliative perspective in all metastatic patients to a more patient and tumor-based approach including targeted and local therapies, allowing treatment with curative intent. With the development of metastasis-directed therapy, such as surgery, local ablation, stereotactic body radiotherapy (SBRT), a new definition of OMD has emerged which considers not only the number of lesions, but also the feasibility of curative treatment of all metastases. Therefore, the definition of OMD does not simply rely on the number of metastatic lesions but also includes the possibility to achieve curative treatment. Finally, repeated imaging is crucial in order to distinguish true OMD from occult systemic metastatic spread in which only a limited number of metastatic sites are detectable and for which systemic treatment is warranted.

All of these specific requirements highlight the major role of imaging in the definition and classification of OMD. Specific items that need to be reported by imaging are the following: the number of organs affected by metastases, the number of metastatic lesions, the number of lesions per organ, the location of metastatic lesions and the relationship with adjacent structures, the tumor growth rate and the metastatic lesions with potential clinical risk. The absolute number of metastatic sites that constitutes the upper limit of the oligometastatic state remains subject of debate, ranging between three to six active extracranial lesions (≤3 in liver and lung parenchyma each) [2]. For lung tumors, nodes are often counted collectively as one lesion. For tumor growth assessment, at least two examinations, performed within an interval of two to six months depending on the tumors’ histological type, are required.

The classification of OMD also varies according to local primary tumor disease control [3]. The term OMD was initially restricted to name newly diagnosed primary tumors with a synchronous limited number of metastases. Closely related terminologies have been developed to refer to other clinical settings including oligo-recurrence for metachronous metastases or new metastases in patients with a prior history of metastatic disease, oligo-progression and oligo-persistent disease for patients treated with systemic therapy, with only few growing metastatic lesions or with few residual metastatic lesions and local disease control, respectively. Oligo-progressive disease has been described in patients with lung cancer treated with targeted therapy or immunotherapy [4,5,6] and is felt to be a consequence of tumor heterogeneity and the development of isolated resistant subclones at only one or a few metastatic sites [7].

In this narrative review, we provide an overview of the different imaging methods for the diagnostic and the pre-therapeutic work up of OMD. We also highlight the role of imaging depending on the primary tumor and the type of treatment.

## 2. Which Is the Best Imaging Method for OMD Diagnostic and Pretherapeutic Evaluation?

Imaging plays a key role in differentiating OMD from PMD. The adequate imaging method for OMD diagnosis, staging and follow-up will be different according to (a) tumor type, (b) timing between measurement and treatment, (c) metastatic location and (d) patient’s individual risk of metastasis. Imaging for assessment of OMD needs to be sensitive—as it is crucial to avoid missing metastasis and ignoring PMD—and specific—as a pathologic confirmation of all lesions is not realistic and acceptable. Consequently, imaging of OMD often requires a combination of targeted imaging for precise loco-regional staging and whole-body assessment. For that purpose, special consideration based on the metastatic risk of each individual situation is mandatory. In general, morphological imaging is usually more useful in early stages, while at later stages, when changes due to treatment are present, the interest of functional imaging including magnetic resonance imaging (MRI) with diffusion weighed imaging (DWI) or positron emission tomography (PET) is growing [8].

### 2.1. Tumor Type

Imaging based on tumor type will be discussed below. Whichever the tumor type, imaging modalities must be selected according to the type of metastatic lesions that are searched for. For instance, while contrast-enhanced (CE) CT is the modality of choice for the follow up of colon cancer, MRI is indicated in patients with mucinous sub-type, as lesions will appear with hyper-intensity in T2 weighted imaging (wi), thus improving the metastatic detection rate in this particular condition.

### 2.2. Timing between Measurement and Treatment

Imaging requirements for OMD evaluation will differ during the course of the disease: at initial diagnosis, at response assessment, and during follow-up. When considering the response assessment of patients with OMD, the standard use of Response Evaluation Criteria in Solid Tumors (RECIST) 1.1 is strongly recommended, as it leads to an objective tumor assessment, which is necessary for adequate follow-up. However, RECIST 1.1 criteria might fail in identifying oligo-progressive disease in which patients develop disease progression in one or a limited number of sites after either a period of stable disease or a partial or complete response. Indeed, RECIST criteria use the sum of a maximum of five target lesions that can hide the progression of an isolated lesion if the other target lesions remain stable. Thus, it is important to consider not only the sum of the target lesions but also the target lesions individually. Some authors have suggested to define oligo-progressive disease as the progression of one to five measurable lesions (according to RECIST 1.1) that are either new or with ≥20% growth of their longest diameter (short-axis in lymph nodes), while other tumor manifestations could shrink or grow less than 20% in diameter [6]. Finally, in oligo-progressive and oligo-persistent disease, the depth and duration of the response to systemic therapy evaluated by imaging using RECIST 1.1 criteria should be considered when assessing treatment options, particularly when evaluating the feasibility of local treatment.

Follow-up imaging after effective treatment is usually performed at three-month intervals during the first year, six-month intervals during the next two years, and then annually. However, this should be adapted to each tumor type, individual risk of recurrence, and patient clinical condition.

### 2.3. Metastatic Location

Each tumor type has a predilection for specific metastatic sites, which will guide the radiological work-up. For instance, a brain MRI is part of the routine-disease extension work-up in small cell lung cancer due to its frequent brain dissemination, while it is performed only in patients with specific symptoms in most of the other tumor types. A good knowledge of each tumor’s typical dissemination pathway is thus mandatory for selection of the most appropriate imaging tool and precise evaluation of images by the radiologist.

### 2.4. Patient Individual Risk

The patient’s individual risk for metastasis must be assessed in order to select the best imaging modality. Low risk patients usually do not need any imaging, while patients at high risk of metastasis might require repeated and extensive imaging (organ-oriented and whole-body imaging). The individual risk for metastasis is assessed by the oncologist by integrating histological, genetic and morphological characteristics of the tumor, as well as imaging data, including functional imaging features, and tumor response to chemotherapy (percentage of tumor shrinkage) [8].

### 2.5. Imaging Modalities

The most common sites of tumor metastasis are the lungs, liver, lymph nodes and skeleton. The first two sites require a dedicated imaging technique (chest CT for the lung and abdominal CT or MRI for liver). The latter two sites are better assessed by whole-body imaging. Thus, a combination of techniques including targeted and whole-body imaging is frequently used in order to cover all potential metastatic sites. Clinical imaging recommendations for each tumor type are important for standardization and adequate patient management.

#### 2.5.1. Contrast Enhanced Computed Tomography (CECT)

Thoraco-abdomino-pelvic (TAP) CECT is the cornerstone cross-sectional imaging modality for morphological imaging. It is fast, available and widely used in practice in oncologic imaging. CT remains the best imaging tool for lung metastasis, the most frequently involved organ. However, its performance for the diagnosis of pathologic lymph nodes and bone lesions is limited (57% sensitivity for lymph nodes assessment and 72% for bone metastasis) [9,10]. Its sensitivity is also limited for the assessment of pelvic tumors. Consequently, CECT is not appropriate in genital organs tumors, for which MRI and positron emission tomography (PET) will be preferred. The use of TAP CECT is limited by the nephrotoxicity of the iodinated contrast agents and by radiation exposure.

#### 2.5.2. Magnetic Resonance Imaging (MRI)

MRI offers high-sensitivity for lesion detection due to high soft tissue contrast, and is used for local tumor staging, particularly for the assessment of adjacent organ invasion. It is recommended in the routine clinical work-up of patients with genital organs, rectum, head and neck and breast cancers. MRI is the imaging modality of choice for the evaluation of brain metastasis and is indicated in the case of bone metastasis involving the rachis in order to assess potential spinal compression [11]. It is used for the assessment of local lymph node involvement. However, the general rule of 10 mm in short axis is not accurate for local lymph node assessment, as it lacks specificity, particularly in diffusion-weighted imaging (DWI) [12,13]. Consequently, additional morphological criteria are used, for instance in rectal cancer, including round shape, irregular borders and heterogenous signal intensity [14]. The use of MRI is limited by its restricted availability and by the long length of time required for examination, which patients may poorly tolerate.

Apart from the use of MRI for local staging, the use of whole-body MRI imaging is growing, including T1 wi, T2 wi and DWI sequences that have already demonstrated high sensitivity for lesion detection in prostate, breast and melanoma cancers [15,16,17]. Reported sensitivity ranges between 84 and 95% and specificity between 87 and 95%. The use of whole-body MRI is currently limited by MRI availability, but is progressively included in clinical guidelines as a modality to be considered whenever possible.

#### 2.5.3. Metabolic and Receptor-Specific Imaging

These techniques include bone scintigraphy and PET/CT. In the context of OMD, these very sensitive techniques are often required in order to exclude occult metastases. Bone scintigraphy is a sensitive modality for the detection of bone lesions as it relies on bone-seeking radiopharmaceuticals, which accumulate in the remodeling bone. It is currently used for staging and treatment response assessment in intermediate and high risk prostate cancer patients, as bone scintigraphy is known to be sensitive for the detection of osteoblastic lesions [18]. PET/CT can be acquired with different radiopharmaceuticals. The most widely used PET radiopharmaceutical is a fluorinated glucose analogue, ^18^F-fluorodeoxyglucose (^18^F-FDG), which accumulates in lesions with high metabolic activity. PET/CT allows whole-body functional imaging, which is very useful for assessing lymph nodes and distant metastasis. ^18^F-FDG PET/CT has largely replaced bone scintigraphy for breast and lung cancer staging, as it is more sensitive for lytic lesions and brings additional information on other organ involvement [19]. ^18^F-FDG PET/CT usage is limited by its high cost, restricted availability, radiation exposure and limited spatial resolution for lesions smaller than 1 cm. The sensitivity of this technique depends on tumor type and the location of metastasis. For instance, ^18^F-FDG PET/CT is the recommended imaging modality for the evaluation of patients with melanoma and head and neck cancer, while it has little value in the evaluation of patients with clear cell renal carcinoma, for example, due to its low metabolic activity.

PET receptor specific imaging plays an increasing role in the management of certain tumors such as prostate cancer (with prostate-specific membrane antigen—PSMA imaging) or neuro-endocrine tumors (with somatostatine receptor imaging). In these tumors, PET receptor specific imaging is becoming a standard of care for the assessment of disease extension due to its very high sensitivity and specificity. Its use is mainly limited by high cost, radiation exposure and restricted availability. Furthermore, the absence of targeted receptor expression by the tumor may result in false negative results.

#### 2.5.4. Combined Modalities

Interest in a combination of morphological and functional modalities is growing as it allows targeted evaluation of selected organs and whole-body imaging. To this end, combined TAP CECT and ^18^F-FDG PET/CT is of particular interest in many cancers for which both morphologic and metabolic imaging are useful.

## 3. Imaging Work-Up Depending on the Type of Treatment

Although surgery is the preferred treatment of OMD, an increasing number of minimal invasive local therapies have emerged. Imaging work-up may also vary according to treatment options.

Pre-operative surgical work-up must include a high-resolution morphological imaging technique, which explains why TAP CECT remains the reference imaging modality. Surgery-oriented imaging evaluation needs to focus on the size and location of metastases, their relationship with adjacent organs and vessels, and the description of anatomic variations.

The SBRT of solitary metastasis, metastasis ≤ 3 cm and metachronous metastasis has been shown to be independently associated with favorable overall survival [20]. The ideal lesions for SBRT in the lung should be distant from central thoracic structures, including the proximal bronchial tree, esophagus and spinal cord and other radio-sensitive structures such as the stomach, bowel, heart, brachial plexus, liver and spleen. The ease of tumor delineation on imaging is also important, and ideally the tumor edges should be well visualized on CT and not obscured by atelectasis or other intercurrent diseases. In some cases, ^18^F-FDG PET/CT may help in tumor delineation for treatment planning. For spinal metastases SBRT, a decision framework including neurologic (N), oncologic (O), mechanical instability (M) and systemic disease and medical comorbidities (S) parameters, the so-called NOMS framework, was developed by Memorial Sloan-Kettering to select patients for appropriate treatment [21]. Spinal MRI is the imaging modality of choice to rule out local treatment contraindications including spinal cord or bone compression, for which surgical decompression is more appropriate, and spinal instability for which surgical fixation should be considered. Lesions larger than 5 cm and multiple levels of involvement are also relative contraindications.

Local ablative treatment, thermal-ablation (radiofrequency and microwave ablation) or cryo-ablation are an efficient alternative to surgery, in particular for the treatment of liver metastases. Liver MRI is the imaging modality of choice when considering percutaneous thermal ablation of liver tumors. As for surgery, the size and the location of the metastases are the most important parameters for patient selection. A size threshold of 3 cm is recommended for curative treatment. Other specific issues when assessing percutaneous thermal ablation of liver tumors are the relationship between the liver lesion, bile ducts and major vessels and the proximity to hollow viscera such as the stomach or the right colon. Finally, due to a high risk of postoperative liver abscesses, lesion location at less than 5–30 mm from a site of enterobiliary anastomosis is also a common contraindication for thermal-ablation.

## 4. Imaging Work-Up Depending on the Primary Tumor

The optimal imaging combination also depends on the more likely spatial distribution and frequency of metastases that differ between cancer types [22]. The imaging modalities according to primary tumor origin are further described and summarized in Table 1.

### 4.1. Lung Cancer

The concept of OMD in non-small cell carcinomas (NSCLC) has been integrated into the eighth edition of the staging system, recognizing that different sites and numbers of metastases are associated with different prognoses. Malignant pleural effusions or isolated contralateral lung metastases are considered M1a, a single site of extra-thoracic metastatic disease is considered M1b, and more extensive extra-thoracic metastatic disease is considered M1c [23]. Local treatment of OMD in lung cancer showed improved long-term survival, especially in patients with single brain or adrenal gland metastasis and no lymph node involvement [24].

NSCLC is probably the primary tumor for which the definition of OMD varies the most across guidelines and clinical trials, with a maximum number of metastatic sites ranging from three to six, with a maximum of three in the liver and lung, and a maximum total number of metastatic lesions ranging from three to five [25]. Whether mediastinal lymph node involvement should be considered or not as OMD is also a subject of debate, with 25% of experts considering that mediastinal lymph node involvement should not be classified as OMD [26].

A meta-analysis of 757 patients having one to five synchronous or metachronous NSCLC metastases found that most oligometastases were either in the brain (35.5%) or lung (33.6%), followed by adrenal gland (13.0%), bone (8.5%), other (7.8%), liver (2.4%), and lymph nodes (2.4%) [24]. To accurately stage NSCLC, both the European Society of medical Oncology (ESMO) and National Comprehensive Cancer Network (NCCN) guidelines recommend the performing of a brain MRI (or a CECT if contraindicated), a TAP CECT and a ^18^F-FDG PET/CT.

The strengths of ^18^F-FDG PET/CT in lung cancer staging is its higher accuracy for the assessment of nodal and bone metastases over conventional CECT. A meta-analysis of 56 studies evaluating the diagnostic value of ^18^F-FDG PET in patients with NSCLC showed a significantly higher performance of ^18^F-FDG PET/CT over CECT with pooled sensitivities and specificities of 72% and 91%, respectively, in determining mediastinal nodal staging, 77% and 95% respectively for detection of all extrathoracic metastases, and 91% and 98% respectively for bone metastases [27]. The NCCN guidelines do not recommend routine bone scintigraphy for NSCLC staging [28]. The main limitation of ^18^F-FDG PET/CT is its low sensitivity for the detection of brain metastases, justifying the need for additional brain MRI. False-negative findings at ^18^F-FDG PET/CT can be the result of a small lung nodule (<1 cm), breathing artifact, low cellular density in lesions such as in lepidic adenocarcinomas, or low tumor avidity for FDG such as in well differentiated neuroendocrine tumors. Compared to CECT, ^18^F-FDG PET/CT has also demonstrated higher prognostic impact with a five-year survival rate after preoperative evaluation by FDG-PET/CT showing no distant metastasis of 58% versus 33% for conventional CECT [29].

### 4.2. Colorectal Cancer

The ESMO has defined oligometastatic colorectal cancer as disease involving up to two or occasionally three sites including liver, lung, peritoneum, lymph nodes and ovary, with five or sometimes more metastases accessible to loco-regional treatment [30]. Due to unfavorable prognosis, patients with other metastatic sites such as bone and brain are excluded from OMD. A recent European consensus [31] has proposed the inclusion of two more criteria in the definition of OMD, corresponding to common conditions accessible to curative treatment by loco-regional procedures. These criteria are the highest diameter of metastases (up to 3 cm) and the maximum number of lesions per organ (up to three). In addition to these criteria used to define OMD, Pitroda et al. [32] suggested that the integration of molecular subtyping is useful to identify patients with most favorable survival outcomes in order to further define curable oligometastatic colorectal cancer.

TAP CECT is the reference imaging modality for colorectal cancer staging that allows for the exploration of all major sites of metastases (Figure 1). Liver MRI, with injection of hepato-specific contrast agent, is associated with the highest diagnostic accuracy for liver metastases, in particular for small lesions <1.5 cm, and should also be included in the staging work-up [33]. A rectal MRI is mandatory in rectal cancer for local staging [34]. Finally, ^18^F-FDG-PET/CT has shown to improve the selection of patients with colorectal cancer and low tumor burden by depicting extrahepatic and extrapulmonary metastases [35]. In a meta-analysis of 1059 patients, ^18^F-FDG-PET/CT evidenced a lower sensitivity than MRI and CT for liver metastases detection (66%, 89% and 79% respectively) [36]. However, findings from ^18^F-FDG-PET/CT resulted in a change in management in 24% of patients due to the detection of additional extrahepatic disease.

### 4.3. Breast Cancer

There is currently no universally accepted definition of OMD in breast cancer. Approximately 20% of all metastatic breast cancers are considered oligometastatic with a limited number and sites of metastases and can be treated with curative intent [37]. In breast cancer, anatomical sites for metastases are widespread and usually include lymph nodes, bone, liver, lung and brain [8].

An initial imaging evaluation of breast tumors consists of mammography and breast and axillary ultrasound. Breast MRI is indicated for local staging and treatment planning. In patients with increased risk for metastatic dissemination due to locally advanced cancer and/or aggressive histological tumors such as a triple negative breast cancers, the standard staging work-up consists of TAP CECT and ^18^F-FDG PET/CT.

Lymph nodes are assessed by ultrasonography including fine needle aspiration or core needle biopsy if suspicious lymph nodes are identified. Usually, breast metastases spread first to axillary lymph nodes then to infraclavicular (N3a), internal mammary (N3b) and supraclavicular (N3c) regions. Involvement of these locations must be ruled out, as it implies an N3 stage according to TNM classification, which is often considered as inoperable at presentation if no neoadjuvant treatment is offered to reduce tumor burden. With no or few associated metastases, N3 stage disease is still considered as oligometastatic and potentially curable with neoadjuvant chemotherapy and reassessment for surgery.

Axial and proximal appendicular skeleton are the most common sites for bone metastasis in breast cancer and can be of osteoblastic, osteolytic or mixed nature. ^18^F-FDG PET/CT has progressively replaced bone scintigraphy in patients with advanced breast cancer, due to its higher sensitivity for bone metastasis detection but also its ability to detect extra-osseous lesions [38,39] (Figure 2). Minamimoto et al. showed in a prospective analysis that ^18^F-FDG PET/CT has a sensitivity of 93.6% in the detection of bone metastases in breast cancer versus 53.2% with bone scintigraphy [40].

Whole-body MRI is an emerging alternative for the detection of mainly bone, liver and brain metastases. It has the advantage of combining morphological (T1wi/T2wi) data, high spatial resolution and tissue contrast with functional data (DWI and dynamic contrast), allowing the detection of hypercellular tumors. Furthermore, osteolytic lesions in the axial skeleton might present an extension to adjacent soft tissue and result in invasion of the epidural space and a compression of the spinal cord, which can be properly assessed by MRI. It also enables the distinction between osteoporotic fractures and pathological ones.

Depending on the histology, the incidence of lung metastasis can reach up to 40%, particularly in cases of triple-negative breast cancer [41]. For this reason, patients with positive lymph nodes and high risk of distant metastasis may undergo chest CT in order to rule out pulmonary metastasis. Alternatively, ^18^F-FDG PET/CT is also a valuable option. However, its sensitivity in detecting small lung metastases (<1 cm) has shown to be lower when compared to lung CT [42]. The addition of a deep-inspiration breath-hold CT acquisition to the conventional PET/CT acquisition may be a useful alternative to better depict small lung metastases.

MRI with liver-specific contrast agents is the most sensitive technique compared to other imaging modalities, including TAP CECT and ^18^F-FDG PET/CT, for the detection and staging of liver involvement [43]. MRI is of interest in patients with OMD, allowing a precise evaluation of liver tumor burden in view of further treatment planning.

Brain metastasis occurs in about 0.4% of patients at initial diagnosis and in 8% when other extra-cranial metastases are present, appearing most commonly late in the clinical history [43]. Brain metastases are more common in patients with HER2+ and triple negative breast carcinoma. Currently, no guideline recommends brain and medullary imaging at initial presentation in asymptomatic patients. Brain and/or medullary contrast-enhanced MRI (or CT if contraindicated) is then performed only in symptomatic patients.

To sum up, patients who may benefit from screening for OMD are those with initial positive lymph nodes, high-risk histology and advanced local disease. The main techniques for breast cancer staging including TAP CECT and ^18^F-FDG PET/CT. A brain MRI should be carried out only in symptomatic patients. Whole-body MRI is an interesting alternative but needs additional liver specific sequences in order to rule out hepatic metastases.

### 4.4. Prostate Cancer

The diagnosis of OMD in prostate cancer is of interest as focal ablative therapies can be applied in this specific population. These therapies include surgery, SBRT and focal thermal ablation. Local treatment of OMD allows the delaying of the introduction of systemic therapy and the development of further metastatic disease [44,45]. The Advanced Prostate Cancer Consensus Conference (APCCC) defined OMD in prostate cancer as the presence of ≤3 bone or lymph node metastases [46].

Prostate cancer typically metastasizes in lymph nodes and skeleton. Conventional imaging is known to underestimate metastatic disease extension [47]. Traditionally, standard imaging for prostate cancer remains bone scintigraphy for bone metastases and TAP CECT or MRI for lymph nodes and visceral lesions [48]. However, imaging paradigms are changing with the development of alternative modalities. Indeed, while bone scintigraphy and TAP CECT are excellent at discriminating progression versus non-progression, their diagnostic performance for tumor staging (PMD versus OMD), recurrence detection and for assessment of treatment response is limited.

Alternative imaging techniques include choline PET/CT, Prostate-specific membrane antigen (PSMA) PET/CT (Figure 3) and whole-body MRI. ^18^F/^11^C PET/CT radiolabeled choline is a precursor for cell membrane phosolipid synthesis. Choline PET/CT has long been used for restaging in patients with biochemical recurrence and a PSA concentration > 1 ng/mL [48]. It is now progressively replaced by PSMA PET/CT due to its higher diagnostic performance [49]. ^68^Ga/^18^F-PSMA PET/CT is highly sensitive for the detection of prostate cancer lesions (including for lymph nodes as small as 3 mm). It is now recommended in any case of biochemical recurrence after prostatectomy (PSA > 0.2 ng/mL) [50]. PSMA is overexpressed in prostate cancer cells compared to normal prostate tissue and its expression increases with increasing Gleason score. However, it is not expressed in neuro-endocrine or undifferentiated cancer subtypes so it cannot be used in these conditions. As for choline PET/CT, PSMA PET/CT is recommended for restaging in patients with biochemical recurrence and a PSA concentration > 1 ng/mL [48] or a fast PSA doubling time. Additionally, its interest in initial patient staging is growing due to its high sensitivity and specificity for metastases detection, as well as its high inter-reader agreement [51,52]. PSMA PET/CT cannot be used to monitor treatment response, as its performance is altered by the use of hormonotherapy. Whole-Body MRI allows mapping of disease extension with additional information on the risk of neurologic complications due to medullary compression by spinal metastases [15]. Inter-reader agreement is also excellent for this technique.

As less than 10% of newly diagnosed prostate cancer in Western countries are metastatic at initial presentation, imaging of prostate cancer is based on patients’ individual risk. While the definition of risk and the imaging modalities used vary among expert societies, all agree that low risk patients do not require any imaging. MRI is recognized as the modality of choice for local and regional lymph node staging. Imaging assessment of distant metastases currently includes bone scintigraphy or PSMA PET/CT. PSMA PET/CT evidenced higher sensitivity compared to bone scintigraphy for bone metastasis detection, although it remains more expensive and less widely available [53,54]. Whole-body MRI will probably play an important role in staging in the near future [55]. Imaging guidelines for biochemical recurrence depend on the patient’s PSA level and type of prior treatment (prostatectomy vs. local therapy). The European Association of Urology (EAU) recommends no imaging for PSA levels < 1 ng/mL [48]. As PSA rises, PSMA PET/CT is recommended after prostatectomy with an additional value of prostate MRI in patients treated by radiation/local ablation [56]. Choline PET/CT is a valuable alternative when PSMA PET/CT is not available. Considering the particular condition of castration-resistant prostate cancer (defined as a PSA rise or a radiological progression in patients with castration levels of testosterone) [56], imaging assessment should include bone scintigraphy and TAP CECT (instead of PSMA PET/CT due to its limited diagnostic performance in this condition) [57]. The benefit of PSMA PET/CT in this particular population is essentially the selection of candidates for Lu177-PSMA therapy.

In summary, the imaging of prostate cancer includes loco-regional assessment by prostate MRI. Distant metastases may be assessed by PSMA PET/CT or bone scintigraphy and TAP CECT when PSMA PET/CT is not available. The choice of imaging is defined by the patient’s individual risk of metastasis, stage of the disease and imaging tool availability. The availability of modern imaging tools, including PSMA-PET/CT and whole-body MRI, is increasing, and these modalities will surely be incorporated in all steps of patient management in the near future, particularly when considering OMD, which requires sensitive and specific imaging.

### 4.5. Head and Neck Cancer

The lungs, bones and liver are the most frequent sites of head and neck cancer metastatic dissemination, the lungs being the most frequent site, accounting for 70% to 85% of all metastases [58]. The literature regarding OMD in head and neck cancer is limited and there is no consensus on OMD definition in this specific cancer. MRI is widely recognized as the modality of choice for loco-regional staging with high soft-tissue contrast allowing evaluation of bone marrow infiltration of the skull base, para-pharyngeal spaces, intracranial disease and cervical lymph node involvement [59]. Search for distant metastasis is indicated in locally advanced tumors. ^18^F-FDG PET/CT is the modality of choice for distant metastasis assessment with additional liver MRI in case of clinical/radiological suspicion of liver metastasis (Figure 4). Cervical TAP CECT combined with ^18^F-FDG PET/CT is of interest as it allows for the assessment of all preferential metastasis locations in a single procedure.

### 4.6. Soft-Tissue Sarcoma

Soft-tissue sarcomas (STS) include a group of rare and heterogeneous diseases with more than 50 different types of histology. There is currently no definition of OMD considering STS.

STS have in common an overall predilection for metastasizing to the lung, resulting in the need for focused imaging of this region. An unenhanced chest CT is usually the method of choice to rule out pulmonary metastasis. Nevertheless, some soft-tissue sarcomas, such as rhabdomyosarcoma, angiosarcoma and synovial sarcoma have been reported to develop extra-pulmonary metastasis and to spread to the lymphatic system. Other types of sarcomas can metastasize to the bone, such as myxoid liposarcoma and to the retroperitoneum, such as myxoid round cell liposarcoma. The latter can thus benefit from an additional imaging of the abdomen and pelvis [60]. For the vast majority of STS, lymph nodes, liver and brain metastasis remain rare [61]. For histologies with a metastasizing predilection to the retroperitoneum and lymphatic system, the method of choice is TAP CECT with focused biopsies in case of suspicious findings.

^18^F-FDG PET/CT appears to be less sensitive than chest CT in the detection of pulmonary metastasis of sarcomas as a significant portion of proven infracentimetric metastatic lung nodules are PET negative due to their small size and the poor FDG uptake of some STS sub-types [62].

### 4.7. Melanoma

The majority of patients newly diagnosed with melanoma have limited disease extension and good prognosis. Routine imaging is not recommended for all melanoma patients, as overuse of imaging in low-risk patients, for instance with negative sentinel lymph nodes, has been shown to be inefficient and exposes patients to useless radiation [63]. Few studies exist regarding the treatment of OMD in melanoma patients and thus no consensual definition has yet been made.

The most common sites of distant metastases in melanoma patients are the skin, lung, brain, liver, bone, and intestine [64]. Skin and subcutaneous metastases are more likely to be detected during physical examination than by imaging. Metastasis to lung is often the first site of visceral metastasis. Other sites of metastasis such as bone and intestines occur later in disease progression and are rarely the first metastatic site detected. Unusual metastatic sites such as the genito-urinary tract, spleen and heart have also been described. Therefore, the search for metastases from melanoma should be wide, due to its unpredictable and sometimes unusual metastatic locations. For this reason and due to avid ^18^F-FDG uptake by melanoma lesions, whole-body ^18^F-FDG PET/CT is the imaging modality of choice, and has shown to be more sensitive in detecting metastatic disease than TAP CECT alone [65]. A deep-inspiration breath-hold thoracic CT acquisition should be added to the standard free-breathing whole body acquisition in order to better depict small lung metastases. A tailored morphological examination according to ^18^F-FDG PET/CT suspicious findings, such as liver MRI in cases of suspicious liver lesions or TAP CECT in case of suspicious small-bowel metastases, should be carried out for treatment decision. As melanoma has a predilection for brain metastasis, brain MRI (or CT if contraindicated) should also be performed before local treatment planning of OMD in melanoma patients. The use of whole-body non-contrast MRI is of interest, as primary and metastatic melanoma lesions appear as hyperintense in T1wi due to their melanotic content. This feature along with other sequences, such as DWI, has been recently explored in patients with melanoma, demonstrating a sensitivity of 88% and specificity of 98% in the detection of distant metastases [66]. Although the benefit of this imaging modality in terms of cumulative radiation exposure appears interesting in the follow-up context, its role in OMD work-up is not established.

Although no consensual definition of oligometastatic melanoma has been made yet, imaging plays a key role in determining the tumor burden as well as the possibility of curative localized therapeutic strategy. ^18^F-FDG PET/CT plays a pivotal role in the detection of metastasis from melanoma.

## 5. Future Directions

PET/MRI is of interest as it allows precise local staging and whole-body imaging, which is the cornerstone of OMD imaging. It is of particular interest in cancer located in body locations where MRI is superior to CT [67]. Specific indications include hepatobiliary imaging of colorectal cancer, where liver involvement can be assessed by MRI sequences with liver specific contrast and extra-hepatic metastasis can be detected by ^18^FDG-PET/CT uptake and characterized by MRI. Kang et al. showed a change of management in 22% of patients with colorectal cancer when using PET/MRI data compared to conventional TAP CECT [68]. Regarding prostate imaging, combining prostate MRI to PSMA-PET in one-stop shop imaging evaluation seems interesting. Preliminary research has shown its added-value for both initial staging and re-staging after biochemical recurrence [69]. PET/MRI also evidenced incremental benefit compared to PET or MRI alone in gynecological malignancies, particularly in cervical cancer [70]. Regarding head and neck tumors, PET/MRI is of interest, with results suggesting higher diagnostic confidence for local tumor staging and lymph nodes assessment with contrast-enhanced PET/MRI compared to contrast-enhanced PET/CT [71].

The development of new radiotracers for oncology imaging is constantly changing with some promising radiotracers:For bone lesion detection, ^18^F-NaF PET/CT is a highly sensitive tracer for skeletal abnormalities detection, as it has a similar mechanism of uptake as Tc 99 m diphosphonates, with a faster clearance of soft tissue-background activity and a superior spatial resolution [72]. Although ^18^F-NaF PET/CT is a very sensitive technique, it remains unspecific. A further limitation is that this costly method solely depicts bone metastases and is not able to detect lymph node or visceral metastases, hence its use is not widespread in the clinical routine.^68^Ga-conjugated fibroblast activation protein inhibitor (^68^Ga-FAPI) PET/CT is of particular interest as it allows for the imaging of cancer-associated fibroblasts (CAF) with high FAP expression. FAP is overexpressed in a wide range of tumors, some of which have a low avidity for ^18^F-FDG (i.e., sarcoma). As FAP is a potential target for cancer treatment, ^68^Ga-FAPI PET/CT is a promising tool not only for staging but also for guiding potential FAP targeted treatments [73].^18^F-Fluciclovine is a synthetic amino acid. Upregulation of the transmembrane amino acid transport occurs in different types of cancer cells, such as in prostate and breast cancers. ^18^F-Fluciclovine has been studied in prostate cancer recurrence, although it is less accurate than PSMA PET/CT [74]. In breast cancer, ^18^F-Fluciclovine PET/CT seems promising, although further investigations are warranted [75].Tumor angiogenesis plays an important role in tumor growth and the development of metastases. Radiotracer targeting of the αvβ3 integrin (^68^ Ga NODAGA-RGD) allows for the imaging of tumor angiogenesis, which may lead to additional information on tumor biology that is potentially useful to guide treatment decisions [76].Recently, a new tracer targeting CD8+ leukocytes in oncological patients has been developed. This radiotracer is a promising tool to predict early response to immunotherapy [77].The folate receptor is overexpressed in several epithelial cancers such as ovarian, endometrial, renal, breast, lung, colon and prostate carcinomas. The folate receptor has emerged as a promising target for cancer treatment and therefore imaging of the overexpression of folate receptors in cancer cells may help in orienting treatment [78,79]. A folate receptor PET tracer has recently been used for the first time in patients with lung cancer, and future perspectives seem encouraging [80].

## 6. Conclusions

Imaging plays a key role in differentiating OMD from PMD and consequently selecting patients who can potentially benefit from a curative treatment. Radiologists should be aware of this intermediate state between local tumor spread and PMD in order to help clinicians in the diagnosis and staging of OMD. The adequate imaging method for OMD diagnosis, staging and follow-up will differ according to tumor type, timing between measurement and treatment, metastatic location and the patient’s individual risk of metastasis. Imaging of OMD requires a combination of targeted imaging for precise loco-regional staging and whole-body assessment to detect distant metastases. For this purpose, multimodal imaging is often crucial to assess the extent and site(s) of disease in patients at risk of metastatic disease. Although the definition of OMD remains poorly defined for several cancers, the interest for this topic is growing as new therapeutic strategies allow potential curative treatment options in patients with OMD.

## Figures and Tables

**Figure 1 cancers-14-01427-f001:**
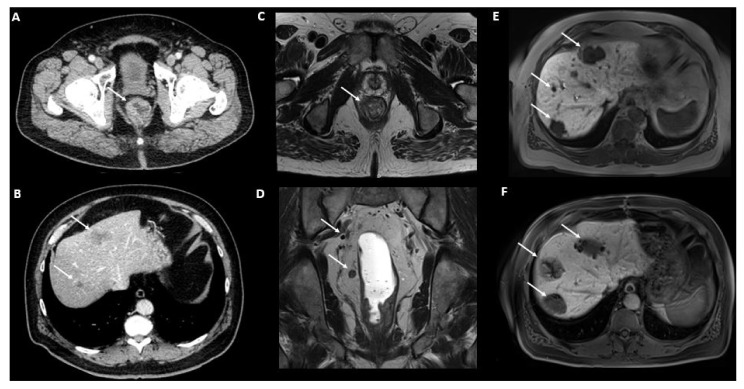
77-year-old patient presenting with constipation and weight loss, a TAP CECT performed shows a rectal mass (**A**, arrow) associated with two liver hypodense lesions in segments II/IVa and VII suspicious of metastases (**B**, arrows). No other metastatic site was identified. Rectal MRI for local staging evidences a cT4 cN2 lower rectal tumor invading ipsilateral levator-ani muscle as shown on axial T2 weighted imaging (**C**, arrow). Coronal T2 weighted imaging evidenced multiple lymph nodes involving the right mesorectum and the right internal iliac region (**D**, arrows). Liver MRI with hepato-specific contrast agent evidenced three liver metastases respectively involving segments II/Iva, VII and VIII (**E**, arrows). The patient was considered as oligometastatic and underwent neo- and adjuvant radiochemotherapy, rectal low anterior resection, surgical wedge resection of segments II/IVa, VII and radiofrequency of segment VIII as shown on post-treatment liver MRI (**F**, arrows) with no recurrent disease at eight years after diagnosis. Abbreviations: TAP: Thoraco-abdomino-pelvic; CECT: contrast-enhanced CT.

**Figure 2 cancers-14-01427-f002:**
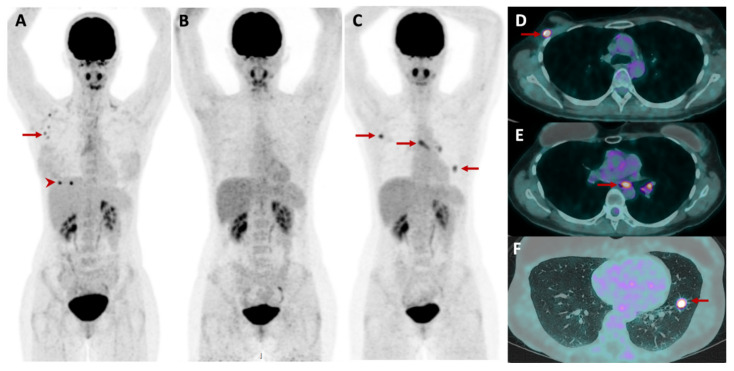
33-year-old patient with advanced NST breast carcinoma. The maximum intensity projection at initial diagnosis (**A**), showed two breast lesions with focal FDG uptake (arrowhead) and pathological radiotracer uptake by axillary lymph nodes (arrow). The patient was in complete remission after treatment by surgery, chemotherapy and adjuvant radiotherapy (**B**), followed by of hormonotherapy. Five years later the patient relapsed (**C**, arrows) with new metastatic lesions, as shown by the fusion images, in the right axilla (**D**, arrow), in mediastinal and left hilar lymph nodes (**E**, arrow) and in the lung (**F**, arrow). The patient was further treated by chemotherapy and local therapy of the metastatic sites: surgery of the right axilla and radiotherapy of the lymph nodes and pulmonary lesion, considered as an oligometastatic recurrence.

**Figure 3 cancers-14-01427-f003:**
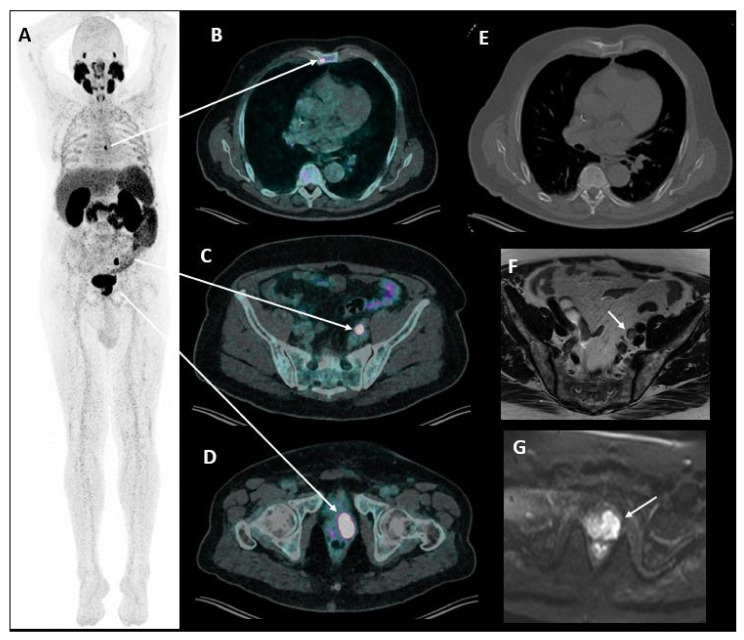
73-year-old patient with suspected prostate cancer due to PSA rise. PSMA PET/CT (**A**–**E**) and prostate MRI (**F**,**G**) evidenced oligometastatic prostate cancer with two metastatic locations including a focal tracer uptake in the sternum (**B**) without correlation on CT (**E**) and one pathologic left external iliac lymph node with focal tracer uptake (**C**) and enlargement seen on T2 weighted imaging (**F**). Note the tracer uptake of the primary tumor on the left prostatic lobe (**D**) with corresponding restricted diffusion on high b-value (**G**).

**Figure 4 cancers-14-01427-f004:**
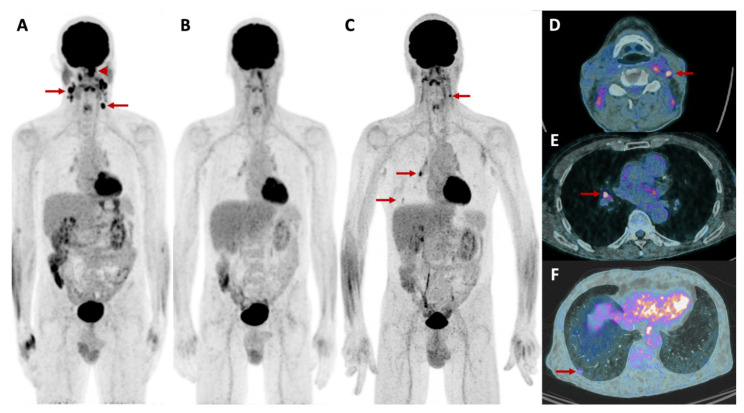
61-year-old patient with a squamous cell carcinoma of the nasopharynx. The maximum intensity projection at initial diagnosis (**A**), showed the nasopharyngeal lesion with focal FDG uptake (arrowhead) and pathological radiotracer uptake by cervical lymph nodes (arrows). The patient was in complete remission after radio-chemotherapy (**B**). Six months later the patient presented an oligo-recurrence (**C**, arrows), as shown by the fusion images, with local left cervical lymph node recurrence (**D**, arrow), and new metastatic lesions in a right hilar lymph node (**E**, arrow) and in the lung (**F**, arrow).

**Table 1 cancers-14-01427-t001:** Imaging recommendations for oligometastasis disease.

Cancer Type	Local Staging	Distant Metastasis Assessment	Alternatives	Comments
Lung	CECT	Brain MRI, TAP CECT, ^18^F-FDG PET/CT	Brain CECT in case of MRI contraindication	Applicable for NSCLC only
Colorectal cancer	Colon cancer: CECT. Rectal cancer: MRI	Liver MRI and TAP CECT	^18^F-FDG PET/CT	
Breast cancer	Mammography, ultrasound, breast MRI	TAP CECT and ^18^F-FDG PET/CT	combined TAP CECT and ^18^F-FDG PET/CT	
Prostate cancer	MRI	Bone scintigraphy or PSMA PET/CT	PET/CT choline when PSMA PET/CT not available, whole-body MRI	TAP CECT in case of castration-resistant prostate cancer
Head and Neck cancer	MRI	TAP CECT and ^18^F-FDG PET/CT	combined TAP CECT and ^18^F-FDG PET/CT	
STS	MRI	thoracic native CT	TAP CECT in specific histologies	

Abbreviations: TAP: thoraco-abdominopelvic; CECT: contrast enhanced computed tomography; STS: soft tissue sacromas.

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
