# Peer review of "Imaging of Oligometastatic Disease"

_cancers, 2022, doi:10.3390/cancers14061427_

Round 1

Reviewer 1 Report

This review article, "Imaging of oligometastatic disease" highlights the role of various currently available imaging modalities in the detection of oligometastatic disease in several commonly encountered solid cancers in the field of oncology.  It provides clinicians and readers with a comprehensive overview of the many diagnostic techniques available in this field. In addition, the studies referenced in this review are appropriately presented and discussed. Thus, I think the article is clinically useful.

Nevertheless, I have following comments:

1. Line 133: PET is firstly mentioned and not in line 155.  

2. Regarding the topic "Future directions": Please add an additional point referring to the role of folate receptor-targeting PET images in ovarian, breast, renal, and lung cancers.  

Author Response

Thank you for these meaningful comments.

Please find our answers:

  1. Line 133: PET is firstly mentioned and not in line 155.  

Answer: Thank you for highlighting this, the change was made accordingly.

  1. Regarding the topic "Future directions": Please add an additional point referring to the role of folate receptor-targeting PET images in ovarian, breast, renal, and lung cancers.  

Answer: Thank you for this suggestion, the role of folate-targeting PET/CT was added in the future directions section.

Reviewer 2 Report

Please see the file attached

Author Response

Thank you for these meaningful comments.

Please find our answers:

  • Methods are missing: as it is a review (also not systematic, but narrative), a brief explanation of the modalities of studies selection is needed (PRISMA guidelines).

Answer: This manuscript is not a systematic review nor a meta-analysis. Thus, PRISMA guidelines are not relevant for the present work.

For clarification, a sentence has been add at the end of the first paragraph (page 2, lines 63-65): “In this narrative review, we provide an overview of the different imaging methods for the diagnostic and the pre-therapeutic work up of OMD. We also highlight the role of imaging depending to the primary tumor and the type of treatment”

Moreover, according to author recommendation we modified the title of the first paragraph from “ Role of imaging in OMD and classification” to “Introduction” (page 1, line 26)

  • Page 8 lines 377-378 and then lines 395-396: differently from choline, PSMA-PET can be used in patients with suspected disease progression or with biochemical relapse with PSA 0.2 ng/mL”. Please see “E-PSMA: the EANM standardized reporting guidelines v1.0 for PSMA-PET” DOI: 10.1007/s00259-021-05245-y. or “EAU-EANM-ESTRO-ESUR-SIOG Guidelines on Prostate Cancer—2020 Update. Part 1: Screening, Diagnosis, and Local Treatment with Curative Intent”. DOI: 10.1016/j.eururo.2020.09.042, for example.

Answer: thank you for this meaningful comment. The corresponding section have been updated including with references:

It is now recommended in any case of biochemical recurrence after prostatectomy (PSA > 0.2 ng/mL).”

Minor comments:

Answer: thank you for highlighting these changes that will improve the quality of the manuscript. We all took them in consideration and made the following suggested changes:

  • Page 2, lines 61-62: you have numbered tumor type, timing between measurement and treatment etc with (1), (2) and so on: in my opinion you should change this numbering not to confuse it with the bibliography list, for example (a), (b) etc, or (A), (B) etc, or (I), (II), etc.

Answer: the modification was done accordingly.

  • Page 4, line 164 you cited (18F-FDG): I think it is better that it is spelled out in full 18Ffluorodeoxyglucose ( 18F-FDG).

Answer: the modification was done accordingly.

  • You often cited 18F-FDG PET: as PET is now always associated with CT, it is not correct, in my view, to name only PET, I suggest to always use 18F-FDG PET, in particular in: a) Page 4, lines 170, 184 b) Page 6, lines 283, 285, 287 c) Page 10 lines 459, 463, 477 (add “18” to “FDG PET/CT”)

Answer: We are now using 18F-FDG PET/CT all along the manuscript.

  • Page 10 line 500: please use PET/CT also for “18F-NaF PET”

Answer: We are added CT as suggested.

  • Page 8, line 354 please add “,” to “surgery SBRT”

Answer: the comma was added.

  • Page 11, line 508 please add “6 ” a “8Ga- FAPI PET/CT

Answer: the “6” was added.

Round 2

Reviewer 2 Report

No comments for Authors